# Interactive Effect of Irrigation Volume and Planting Density on Growth and Salt Uptake in Field-Grown Drip-Irrigated *Suaeda salsa* (L.) Pall.

**DOI:** 10.3390/plants12061383

**Published:** 2023-03-20

**Authors:** Qiang Xu, Hongguang Liu, Mingsi Li, Pengfei Li

**Affiliations:** 1College of Water Conservancy & Architectural Engineering, Shihezi University, Shihezi 832000, China; xuqiang@stu.shzu.edu.cn (Q.X.); limingsi@shzu.edu.cn (M.L.); lipengfei@shzu.edu.cn (P.L.); 2Key Laboratory of Modern Water-Saving Irrigation of Xinjiang Production & Construction Group, Shihezi University, Shihezi 832000, China

**Keywords:** *Suaeda salsa* (L.) Pall., drip irrigation, saline–alkali soil, irrigation volume, planting density

## Abstract

Planting halophytes such as *Suaeda salsa* (L.) Pall. under drip irrigation is a viable solution for the remediation of saline soils. We conducted this study to investigate the effects of different irrigation volumes and planting densities on the growth and salt uptake of *Suaeda salsa* under drip irrigation. The plant was cultivated in a field using drip irrigation at various irrigation volumes (3000 m·hm^−2^ (W1), 3750 m·hm^−2^ (W2), and 4500 m·hm^−2^ (W3)) and planting densities (30 plants·m^−2^ (D1), 40 plants·m^−2^ (D2), 50 plants·m^−2^ (D3), and 60 plants·m^−2^ (D4)) to examine the effects on growth and salt uptake. The study revealed that the amount of irrigation, planting density, and interaction between the two significantly affected the growth characteristics of *Suaeda salsa*. The plant height, stem diameter, and canopy width increased simultaneously with an increase in the irrigation volume. However, with an increasing planting density and the same irrigation volume, the plant height first increased and then decreased, while the stem diameter and canopy width decreased simultaneously. The biomass of D1 was the highest with the W1 irrigation, while that of D2 and D3 were highest with the W2 and W3 irrigations. The amount of irrigation, planting density, and their interaction significantly affected the ability of *Suaeda salsa* to absorb salt. The salt uptake increased initially and then decreased with an increasing irrigation volume. At the same planting density, the salt uptake of *Suaeda salsa* with the W2 treatment was 5.67~23.76% and 6.40~27.10% higher than that with W1 and W3, respectively. Using the multiobjective spatial optimization method, the scientific and reasonable irrigation volume for planting *Suaeda salsa* in arid areas was determined to be 3276.78~3561.32 m^3^·hm^−2^, and the corresponding planting density was 34.29~43.27 plants·m^−2^. These data can be a theoretical basis for planting *Suaeda salsa* under drip irrigation to improve saline–alkali soils.

## 1. Introduction

Saline soils occupy a total area of 3.63 × 10^7^ hm^2^ in China, accounting for 4.88% of the available land [1,2]. The largest land area is 1.34 × 10^7^ hm^2^ in Xinjiang, accounting for 19.75% of the available land area and 36.80% of all saline soil areas in China [3,4]. Eighty percent of the saline–alkali land in China has not been developed [5,6], even though underutilized saline–alkali soil is a valuable resource that can serve as a strategic reserve for cultivated land. The rational and sustainable use of saline–alkali soils can lead to better economic and social development and national food security [7,8]. *Suaeda salsa* (L.) Pall. is an annual succulent halophyte with strong salt resistance, and it is widely distributed in northern China [9,10,11,12,13]. Suaeda salsa not only has economic value in forage development and utilization, it is also an effective measure for improving and developing saline soils [14,15,16].

There are two crucial elements for managing cultivated grassland, which are suitable soil water content and planting density. Only when these two conditions are ideal can a steady and high yield be achieved [17]. Some researchers have developed irrigation schemes for *Suaeda salsa* based on their experience. Qi et al. [18] applied an irrigation system in which *Suaeda salsa* is irrigated once every three days with an irrigation amount of 0.67~1.00 m^3^·hm^−2^ in the early growth stage and once every 10~15 days in the later stage, with an irrigation amount of 1.67~2.00 m^3^·hm^−2^. Guo et al. [19,20] divided drip irrigation six times according to the actual water demand, with a total irrigation volume of 3600~3900 m^3^·hm^−2^. Pan et al. [21,22] drip irrigated once every 15 days, 8~10 times over the whole growing period, and the irrigation volume during the two-year experiment was 6600 m^3^·hm^−2^ and 4700 m^3^·hm^−2^, respectively. Zhao et al. [23] applied a high-frequency irrigation system 30 times over the growing period, and the total irrigation amount was 4665 m^3^·hm^−2^. Despite some practical cases, there is still a lack of a rational and scientific irrigation protocol for *Suaeda salsa*. The irrigation methods used were based on an equal time and equal amount of water during the entire growth period. These studies also employed sowing by drilling and high-density planting [20,24,25]. However, research by Shao et al. [26] revealed a strong correlation between the planting density and the growth of *Suaeda salsa*. They found that the plant height reached 120 cm when the planting density was less than ten plants per square meter but reached only 40 cm without branching when there were 1088 or more plants per square meter. Similar findings by Pan et al. [22] showed that the planting density of *Suaeda salsa* under drip irrigation significantly impacted the yield, indicating the need to manage plant density.

In summary, the improvement of saline–alkali soils by planting *Suaeda salsa* is related to their growth state and salt uptake index, which are further affected by the irrigation volume, planting density, and the interaction between the two variables. However, there are few reports on how the irrigation volume and planting density affect the growth of *Suaeda salsa* and its ability to absorb salt under drip irrigation. The goal of this study was to create an irrigation schedule that takes the developmental stages of *Suaeda salsa* into account and focuses on the following: (1) the effects of various irrigation volumes and planting densities on the growth of *Suaeda salsa* under drip irrigation; (2) the effects of various irrigation volumes and planting densities on the salt uptake by *Suaeda salsa*; and finally, (3) the establishment of a theoretical foundation for planting *Suaeda salsa* under drip irrigation to improve saline–alkali soils in arid regions using a scientifically sound irrigation volume and planting density.

## 2. Results

### 2.1. Effects of Different Planting Densities and Irrigation Volumes on Plant Height of Drip-Irrigated Suaeda salsa

Plant height is an essential morphological indicator to measure crop growth. An appropriate plant height is conducive to the rational distribution of the canopy, improving the use of light energy, and improving crop production and yield [27]. Table 1 shows that throughout the growth period of *Suaeda salsa*, the plant height change trend for each treatment was consistent; the growth rate increased rapidly from the seedling stage to the adult stage before reducing as it approached the flowering stage. From the seedling stage, the plant height of each treatment was significantly different (*p* ≤ 0.05), and as the plant developed, the difference among the treatments became even more significant. The heights of the plants treated with W2 and W3 were 18.66~79.19% and 5.21~56.75% higher than that of the plants treated with W1. As the planting density increased, the plant height initially increased and then decreased under each amount of irrigation. Under the condition of irrigation volume W1, the plant height of D2 was the highest; it was 15.19~27.50%, 7.01~24.89%, and 31.90~35.54% higher than that of D1, D3, and D4, respectively. However, in W2, the plant height of D3 was the highest, 37.26~62.30%, 12.25~32.20%, and 43.77~70.36% higher than that of D1, D2, and D4, respectively. In W3, the plant height of D3 remained the highest, 33.38~57.15%, 4.58~14.11%, and 35.18~44.13% higher than that of D1, D2, and D4, respectively. Thus, it can be inferred that the amount of irrigation and planting density significantly affect the plant height of *Suaeda salsa*, and the interaction between them is extremely significant.

### 2.2. Effects of Different Planting Densities and Irrigation Volumes on Stem Diameter of Drip-Irrigated Suaeda salsa

As is evident from Table 2, the stem diameter of each treatment was significantly different (*p* ≤ 0.05), and as the plant developed, the difference among the treatments became even more significant. The stem diameter of the plants treated with W1, W2, and W3 increased simultaneously with the amount of irrigation. The stem diameters of *Suaeda salsa* treated with W2 and W3 were 1.20~88.29% and 0.98~52.17% higher than that of the plants treated with W1. As the planting density increased, the stem diameter decreased gradually under each irrigation amount. Under irrigation volume W1, the stem diameter of D1 was the largest, which was 1.14~2.07, 1.29~3.23, and 1.82~8.57 times more than that of D2, D3, and D4, respectively. The stem diameter of D1 remained the largest under the W2 and W3 irrigation treatments. Under W2, it was 1.05~1.51, 1.52~2.22, and 1.98~5.38 times, while under W3, it was 1.20~1.59, 1.36~2.38, and 1.98~6.07 times larger than that of D2, D3, and D4, respectively. Thus, it can be inferred that the amount of irrigation and planting density significantly affect the stem diameter of *Suaeda salsa*, and their interaction has significant implications.

### 2.3. Effects of Different Planting Densities and Irrigation Volumes on the Canopy Width of Drip-Irrigated Suaeda salsa

Table 3 shows that the development of the canopy width of *Suaeda salsa* throughout the growing season was comparable to that of the stem diameter in each treatment and that the growth rate of the canopy width was relatively uniform from the seedling stage to seed formation. The crown size of each treatment was significantly different (*p* ≤ 0.05), and as the plant developed, the difference among the treatments became even more significant. The canopy width of *Suaeda salsa* treated with W1, W2, and W3 decreased with the increase in irrigation volume. The canopy widths of the plants treated with W2 and W3 were 3.47~102.58% and 2.07~94.83% higher than those treated with W1. With the increase in the planting density, the canopy width decreased rapidly under each irrigation amount. Across all of the irrigation volumes of W1, W2, and W3, D1 had the largest canopy width, showing fairly consistent trends. Under W1, the canopy was 1.25~1.68, 3.26~4.88, and 4.72~8.42 times larger; under W2, it was 1.15~1.75, 2.06~3.52, and 4.25~5.32 times larger; while under W3, it was 1.07~1.75, 2.29~2.96, and 4.17~6.18 times larger than that of D2, D3, and D4, respectively. Thus, it can be inferred that the amount of irrigation and planting density significantly affect the canopy width of *Suaeda salsa*, and their interaction has significant implications.

### 2.4. Effects of Different Planting Densities and Irrigation Volumes on the Biomass of Drip-Irrigated Suaeda salsa

The biomass change trend of each treatment was consistent throughout the growth period of *Suaeda salsa* (Table 4). The biomass increased greatly from the seedling stage to the mature stage and slowed down after the flowering. The biomass of each treatment was significantly different (*p* ≤ 0.05), and as the plant developed, the difference among the treatments became more significant. Under W1, W2, and W3, the biomass of *Suaeda salsa* increased with the increase in the amount of irrigation, except for planting density D1. The biomass of plants treated with W2 and W3 was 12.33~30.28% and 10.14~35.10% higher than those treated with W1.

As the planting density increased, the biomass showed a different trend under different amounts of irrigation. Under the condition of the W1 irrigation, the biomass of *Suaeda salsa* gradually decreased with the increase in the planting density, with the biomass of D1 being the highest, which was 0.48~6.59%, 6.17~14.55%, and 12.97~29.88% higher than that of D2, D3, and D4, respectively. Under W2, the biomass initially increased before declining with an increase in the planting density. The biomass of D2 was the highest, which was 16.27~28.23%, 0.93~8.56%, and 19.50~40.10% higher than that of D1, D3, and D4, respectively. Under the W3 irrigation, the biomass showed a similar trend as under W2. However, here, the biomass of the D3 treatment was the highest, 8.05~17.68%, 3.27~6.81%, and 12.43~26.46% higher than that of D1, D2, and D4, respectively. Thus, it can be inferred that amount of irrigation and planting density significantly affect the biomass of *Suaeda salsa*, and the interaction between them is highly significant.

### 2.5. Effects of Different Planting Densities and Irrigation Volumes on the Ash Content and Amount of Salt Uptake of Drip-Irrigated Suaeda salsa

As shown in Table 5, throughout the growing period of *Suaeda salsa*, the ash content with each treatment initially increased, followed by a decrease from the mature stage to the seed setting stage. The ash content was the highest during the adult stage, dropped significantly during the flowering stage, and was the lowest during the seed setting stage. The ash content of W2 and W3 decreased by 1.47~8.99% and 16.56~21.59%, respectively, compared with the W1 treatment. The ash content showed a negative correlation with the irrigation volume. The ash content of D1 was 8.37~11.58%, 10.01~17.50%, and 14.32~19.80% higher than that of D2, D3, and D4, respectively.

Table 5 shows that the salt uptake of *Suaeda salsa* had two peaks in each treatment throughout the growing period. The ash content was the highest in the mature stage, as was the salt uptake, which accounts for the first peak. Although the ash content of *Suaeda salsa* was the lowest in the fruiting stage, the biomass reached its maximum, and the salt uptake reached the second peak. With an increase in the irrigation, the salt uptake of *Suaeda salsa* initially increased and then decreased. At the same planting density, the salt uptake was highest with W2 (except W1D1), which was higher than that with W1 and W3 by 5.67~23.76% and 6.40~27.10%, respectively. Thus, it can be inferred that the amount of irrigation and planting density significantly affect the salt uptake of *Suaeda salsa*, and the interaction between them is highly significant.

### 2.6. Effects of Different Planting Densities and Irrigation Volumes on the WUE and WSE of Drip-Irrigated Suaeda salsa

The effects of different planting densities and irrigation volumes on the water production efficiency and salt uptake efficiency of *Suaeda salsa* are shown in Table 6. The declines in the WUE and WSE with an increasing irrigation volume show similar trends. The WUE and WSE decreased under the W1 irrigation as the planting density increased, reaching their highest levels under the planting density D1, where they were higher by 6.60% and 18.59%, 14.56% and 32.14%, 29.72%, and 55.46%, respectively, than those in the D2, D3, and D4 treatments. Under the W2 and W3 irrigation volumes, the WUE and WSE first increased and then decreased with an increase in the planting density. However, D2 showed the highest values for WUE and WSE in the irrigation volume W2 (28.29% and 18.05%, 8.53% and 10.56%, 39.95% and 50.96% more than in D1, D3, and D4, respectively), while D3 showed the highest value for WUE and WSE in the irrigation volume W3 (17.50% and 1.96%, 6.82% and 1.96%, 26.34% and 28.40% more than in D1, D2, and D4, respectively).

### 2.7. Multiobjective Optimization of Each Index Based on the Spatial Analysis Method

We used the irrigation volume and planting density as independent variables and each parameter as a dependent variable to obtain a binary quadratic regression equation (Figure 1 and Table 7). We also calculated the maximum values for each parameter (Table 8). With an irrigation volume of 4163.27 m^3^·hm^−2^ and a planting density of 43.47 plants·m^−2^, Table 8 shows that a maximum yield of 20.88 t·hm^−2^ is possible. Similarly, a maximum salt intake of 5547.60 kg·hm^−2^ was feasible with the irrigation at 3612.24 m^3^·hm^−2^ and planting density at 36.12 plants·m^−2^. The maximum theoretical value for WUE was 5.91 kg·m^−3^, possible at an irrigation volume of 3000 m^3^·hm^−2^ and planting density of 39.18 plants·m^−2^. The maximum theoretical value for WSE was 1.74 kg·m^−3^, possible at an irrigation volume of 3000 m^3^·hm^−2^ and planting density of 30 plants·m^−2^. Different irrigation volumes and planting densities resulted in the maximum theoretical values for each parameter.

Farmers are, in fact, more concerned with the yield of *Suaeda salsa*, whereas water resources management organizations and academic researchers are more concerned with salt intake capacity, WUE, and WSE. As a result, this study’s choice of optimization indicators included yield, salt uptake, WUE, and WSE. Therefore, we used the spatial analysis method for multiobjective optimization. The unit sizes of each index were normalized, and the spatial domain of the binary quadratic equation of each index was projected onto the two-dimensional plane to obtain the contour map of the relative values of each index (Figure 2). An extensive analysis of these indices revealed that the highest achievable yield, salt uptake, WUE, and WSE were obtained at an irrigation volume of 3276.78~3561.32 m^3^·hm^−2^ and a planting density of 34.29~43.27 plants·m^−2^ and were 80% of their maximum theoretical values (Figure 3).

## 3. Discussion

Irrigation and density have a two-way regulation effect on individuals and populations of *Suaeda salsa*. Increasing the plant density negatively impacts individual development because of the increase in the population and interplant competition [28]. However, irrigation can significantly promote the growth of *Suaeda salsa* vegetation [29,30]. Therefore, balancing the promotion and control effects of irrigation and planting density is necessary for developing efficient populations [31]. We found that the amount of irrigation and planting density significantly affected plant height, stem diameter, and canopy width, and the interaction between them was highly significant. The plant height, stem diameter, and canopy width of *Suaeda salsa* showed an increasing trend with the increase in the amount of irrigation, and the W2 plants improved more than the W3 plants.

Irrigation promotes the branching and canopy width [32,33], leaf growth and development [34], and photosynthetic capability [35,36] of plants. This results in an increase in the aboveground biomass, which is consistent with the finding that irrigation can increase the biomass of Elymus nutans [37]. Crop growth and development are not only affected by environmental factors, such as irrigation and fertilization [38], but also limited by planting density [39]. Changes in crop planting density affect the interactions between individuals [40], causing density constraints in plant populations [41]. Plant density limits the number of resources obtained by individuals within the population, causing competition for light, water, and nutrition among adjacent plants [42]. Population density can significantly affect the characteristics of individuals within a plant population. This study showed that different plant densities affected the plant height, stem diameter, and canopy width (*p* ≤ 0.05, Table 1, Table 2 and Table 3). With the increase in the planting density, the plant height increased initially and then decreased, whereas the stem diameter and canopy width decreased constantly. Wang et al. [43] found that increasing the planting density resulted in an increased plant height and reduced stem diameter in pepper plants, as dense planting compresses the horizontal growth space of plants, thereby promoting vertical growth. This might be because the planting density can regulate growth and development by affecting physiological conditions and canopy structure [44]. As the planting density increases, adjacent plant canopies trigger a shade avoidance response in plants, allowing them to distribute more assimilates from storage organs to nutrient organs. This results in the rapid vertical elongation of stems, thinner walls, and reduced internode diameter [45]. We discovered that growth in the D4 density was inhibited across the three irrigation volumes in this study, because as the population density increases, individuals maintain growth, inevitably forming a resource competition pattern [40]. An increase in plant density causes fiercer competition among individuals for available light and nutrients, weakens photosynthesis, and decreases biomass [46]. The ability of individual plants to accumulate carbohydrates reduces, growth is blocked, and biomass is reduced [47].

The relationship between plant density and soil moisture is both reciprocal and antagonistic. Excessive plant density accelerates plant transpiration, while a low plant density and greater surface evaporation affect the soil water use efficiency [48]. Consequently, only by coupling plant density and soil moisture can stable crop yields be maintained and the water use efficiency be improved. This is consistent with the results of this study, where an increased irrigation water volume gradually increased the ideal density for obtaining high yields. Under the condition of the W1 irrigation volume, D1 had the highest biomass; under the condition of the W2 irrigation volume, the biomass of D2 was the highest; and under the condition of the W3 irrigation volume, D3 had the highest biomass. The biomass under W1, W2, and W3 increased with an increase in the amount of irrigation. The biomass of *Suaeda salsa* treated with W2 and W3 was higher than that of the plants treated with W1. Feng et al. [37] showed similar results when the yield changed with the change in the amount of irrigation at a fixed planting density. Our results show that the WUE and WSE displayed a decreasing trend with the increase in the irrigation volume. With the W1 irrigation volume, the WUE and WSE decreased with the increase in the planting density. The WUE and WSE of D1 were the highest, 6.60~18.59 and 29.72~55.46% higher than those of D2, D3, and D4, respectively. However, under the W2 and W3 irrigation volumes, the WUE and WSE increased first and then decreased with the increase in the planting density. D2 had the highest WUE and WSE under the W2 irrigation volume, while D3 had the highest WUE and WSE under the W3 irrigation volume. These findings are consistent with those of Zhang et al. [49], who demonstrated that an increase in the planting density causes the water use efficiency to increase initially before decreasing.

By adjusting the mutual regulatory effects of irrigation and density on the individual development and population structure of *Suaeda salsa*, Luo et al. [50] hypothesized that the promoting effect of irrigation and the inhibiting effect of density coordinate the overall benefits to achieve optimally high yields [28]. We obtained similar results in this study. Suaeda salsa displayed the best achievable characteristics when all four parameters—yield, salt uptake, WUE, and WSE—reached 80% of their maximum theoretical values (Figure 3). The ideal irrigation volume to achieve this was 3276.78~3561.32 m^3^·hm^−2^, which is close to the total irrigation amount reported by Guo et al. [19,20]. The ideal planting density was 34.29–43.27 plants·m^−2^, which is similar to the results of Shao et al. [26]. Luo et al. [51] considered that in tall grass, moderately sparse planting is conducive to light penetration and aeration, thus improving the nutrient area of a single clover plant, promoting branch formation, and increasing the yield of sweet clover. *Suaeda salsa* can be planted thinly to maximize the potential of each plant and boost production because of its strong branching ability and high individual production capacity. The current planting method, which uses a high planting density and high amounts of irrigation with drill planting, should be changed to sparse planting with the appropriate amount of irrigation when drip irrigation is used.

## 4. Materials and Methods

### 4.1. Study of the Test Area

The study was conducted at the Bachu County Modern Agricultural Industrial Park (78,55’ E, 39,75’ N, 1072.80 m above sea level) in Kashgar from May 2022 to October 2022. This area has a temperate continental dry climate, with drought and little rain, windy and dusty weather, and a long frost-free period. The region experiences an annual average temperature of 11.80 °C, 4434 h of sunshine, 213 days without frost, 50 mm of average precipitation, and over 2500 mm of average evaporation in a year (see Figure 4 for the meteorological data and geographical location of the study area). Soil was air-dried, crushed, and passed through a 2 mm sieve to remove impurities. The contents of sand, silt, and clay, as determined by the screening and hydrometer methods, were 15.86%, 79.90%, and 4.24%, respectively. The water-holding capacity of the soil was studied using the drainage method. Table 9 contains a complete list of the physical and chemical characteristics of the soil used.

### 4.2. Experimental Design

Based on reference crop evapotranspiration (ET_0_) for the months of May through October 2020 to 2022, the irrigation volumes of 3000 m^3^·hm^−2^, 3750 m^3^·hm^−2^, and 4500 m^3^·hm^−2^ were set, with 40%, 50%, and 60% ET_0_ in each month and denoted as W1, W2, and W3, respectively. The planting densities of *Suaeda salsa* were 30 plants·m^−2^, 40 plants·m^−2^, 50 plants·m^−2^, and 60 plants·m^−2^, respectively, designated as D1, D2, D3, and D4 (see Table 10 for the experimental design). The area of each experimental plot was 50 m^2^, with three replicates.

Before sowing, the experimental area was prepared by leveling the soil. *Suaeda salsa* was sown within 15 cm on both sides of the drip irrigation belt by hand digging and seeding. The plant wide rows were 50 cm and narrow rows were 30 cm, and the seeding rate was set at 30 kg·hm^−2^. The irrigation method was drip irrigation (see Figure 5 for the drip irrigation planting of *Suaeda salsa*), which used groundwater with a salinity of 0.35 g·L^−1^. The seeding was conducted on May 10 using the dry sowing and wet-out method. During this stage, the plots were uniformly irrigated with 500 m^3^·hm^−2^ of water five times to ensure seedling emergence. The seedlings started to emerge on May 30. The postsowing irrigation system was implemented according to Table 11. Compound fertilizer (N:P_2_O_5_:K_2_O = 20:14:6) in the amount of 500 kg·hm^−2^ was applied to the experimental site before sowing. Postseeding fertilizer was administered twice on July 15 and August 25, with 150 kg·hm^−2^ each time.

### 4.3. Data Collection

The sampling of *Suaeda salsa* was conducted on 10 June (seedling stage), 11 July (mature stage), 20 August (blooming stage), and 15 October (fruiting stage). The following data were collected:(1)Plant height: the height from the ground to the tip of the main stem was measured using a measuring tape [52].(2)Stem diameter: the stem diameter was measured at 2 cm above the ground with a Vernier caliper [4].(3)Canopy width: the average width of the plant in the north–south and east–west directions was measured using a measuring tape [4].(4)Biomass (t·hm^−2^): Plant samples for each stage were collected from a random sampling area in each plot of 1 m × 1 m. The processed fresh materials were placed in an oven at 105 °C for 10 min and dried at 80 °C until the weight was constant, providing the biomass [52].(5)Coarse ash content: one gram of dried plant material was placed in a muffle furnace and incinerated for 18 h at 550 °C, placed in a dryer, cooled, and weighed, providing the ash content of the plant [53].(6)Salt uptake amount by pan [21,22]:
S = Ash × Y(1) where S is the amount of salt uptake (kg·hm^−2^); Ash is the coarse ash content (kg·hm^−2^); and Y is the biomass (kg·hm^−2^).

(7)Irrigation water use efficiency by [54]:WUE = Y/W(2) where WUE is the irrigation water use efficiency (kg·m^−3^); Y is the biomass (kg·hm^−2^); and W is the irrigation volume (m^3^·hm^−2^).

(8)Irrigation water salt transfer efficiency:WSE = S/W(3) where WSE is the irrigation water salt transfer efficiency (kg·m^−3^); S is the amount of salt uptake (kg·hm^−2^); and W is the irrigation volume (m^3^·hm^−2^).

### 4.4. Data Analysis

Excel 2003 (Microsoft, Redmond, WA, USA) was used to collate the test data. Each index was repeated three times and averaged, and the data fitting and statistical analysis were performed using SPSS 19.0 (IBM, Armonk, NY, USA). The average values of the parameters in each treatment were calculated, and a correlation analysis was conducted. For multiple comparisons, the Tukey comparison method was employed. Using MATLAB R2023a (Mathworks, Nadik, MA, USA), we further examined the data to determine the relationship between plant density, irrigation volume, and indicators, created a binary regression model, and used the fminsearch function to determine the best theoretical value. The graphs were generated using Origin 2020 (OriginLab, Northampton, MA, USA).

## 5. Conclusions

This study shows that different planting densities and irrigation volumes significantly affected the growth, biomass, and salt uptake of drip-irrigated *Suaeda salsa* (*p* < 0.05). The following conclusions were drawn:(1)The plant height, stem diameter, and canopy width of *Suaeda salsa* were significantly affected by the amount of irrigation, planting density, and the interaction between the two factors. Under the same amount of irrigation, with an increasing planting density, the plant height first increased and then decreased, whereas the stem diameter and canopy width decreased consistently.(2)The amount of irrigation, planting density, and the interaction between them significantly affected the plants’ biomass, WUE, and WSE. With the increase in the irrigation volume, the suitable planting density of *Suaeda salsa* also increased. Under the W1 irrigation volume, the biomass, WUE, and WSE of D1 were the highest; under the W2 irrigation volume, the biomass, WUE, and WSE of the D2 treatment were the highest; while under the W3 irrigation volume, the biomass, WUE, and WSE of D3 were the highest.(3)The amount of irrigation, planting density, and the interaction between them also significantly affected the salt uptake by *Suaeda salsa*. As the amount of irrigation increased, the salt intake of *Suaeda salsa* first increased and then decreased. At the same planting density, the salt intake was higher in the W2 treatment than in the W1 and W3 treatments.

The scientific and reasonable irrigation volume for planting *Suaeda salsa* in arid areas was determined to be 3276.78~3561.32 m^3^·hm^−2^, and the corresponding planting density was 34.29~43.27 plants·m^−2^. These results are based on the multiobjective optimization of the spatial analysis method and provide a theoretical basis for planting *Suaeda salsa* under drip irrigation to improve saline–alkali soils.

## Figures and Tables

**Figure 1 plants-12-01383-f001:**
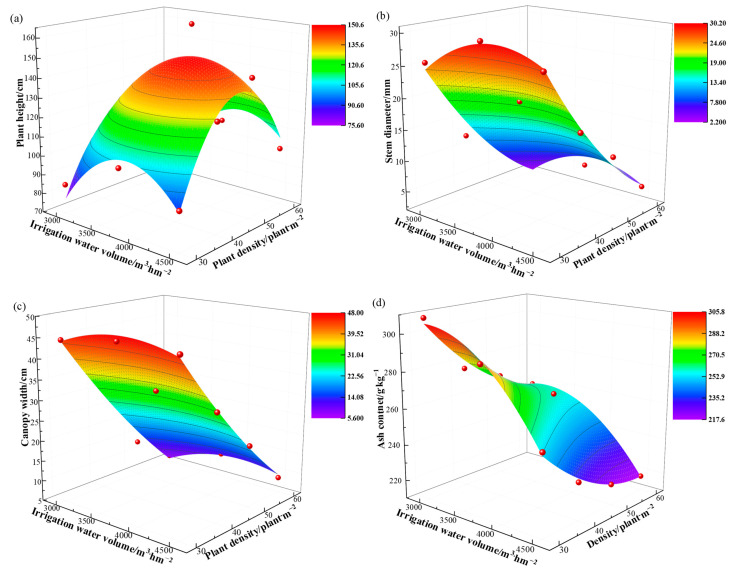
Three-dimensional surface map of the irrigation water and density input for each evaluation indicator. In the figure (**a**–**h**) represent plant height, stem diameter, canopy width, ash content, biomass, salt uptake, WUE and WSE, respectively. The red dots in the figure represent the field observation values of the evaluation indicators.

**Figure 2 plants-12-01383-f002:**
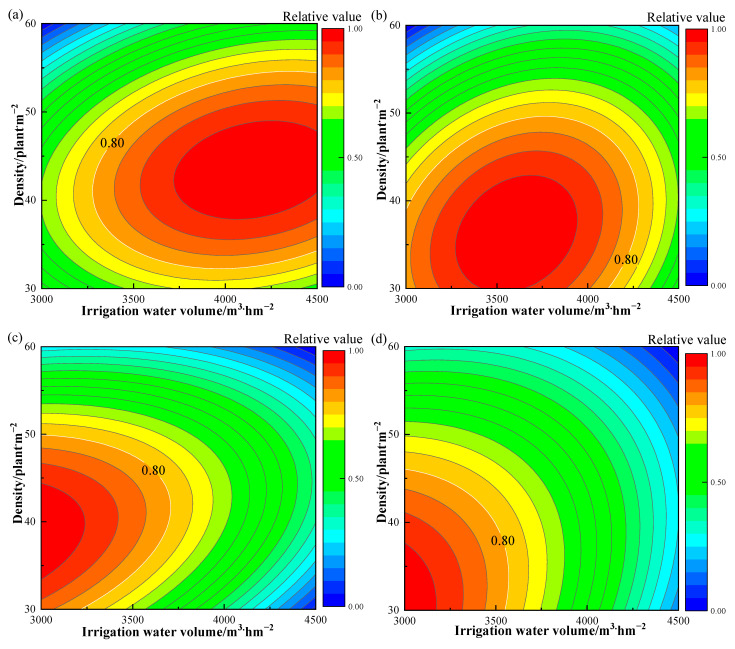
Contour maps of the relationship between each evaluation index at different irrigation volumes and densities. In the figure (**a**–**d**) represent biomass, salt uptake, WUE and WSE, respectively.

**Figure 3 plants-12-01383-f003:**
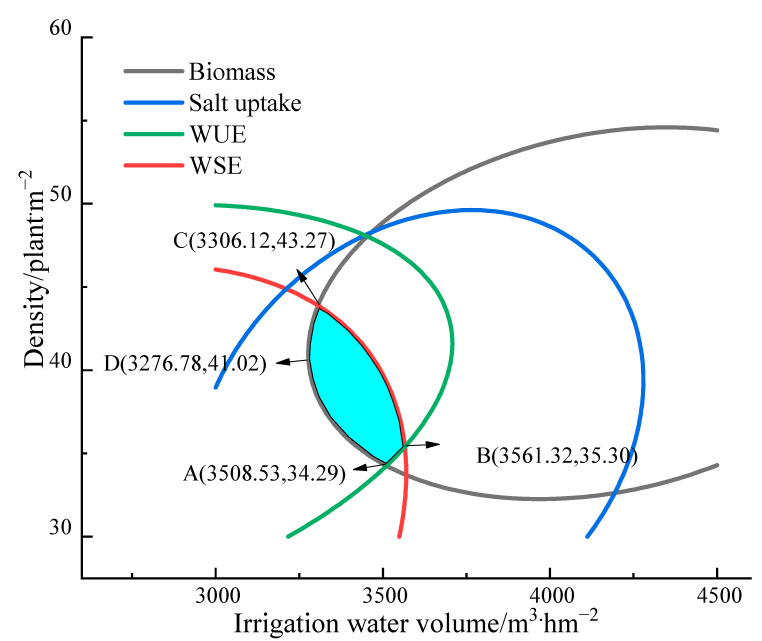
Concurrent evaluation of indices. The blue filled-in area meets the evaluation requirements.

**Figure 4 plants-12-01383-f004:**
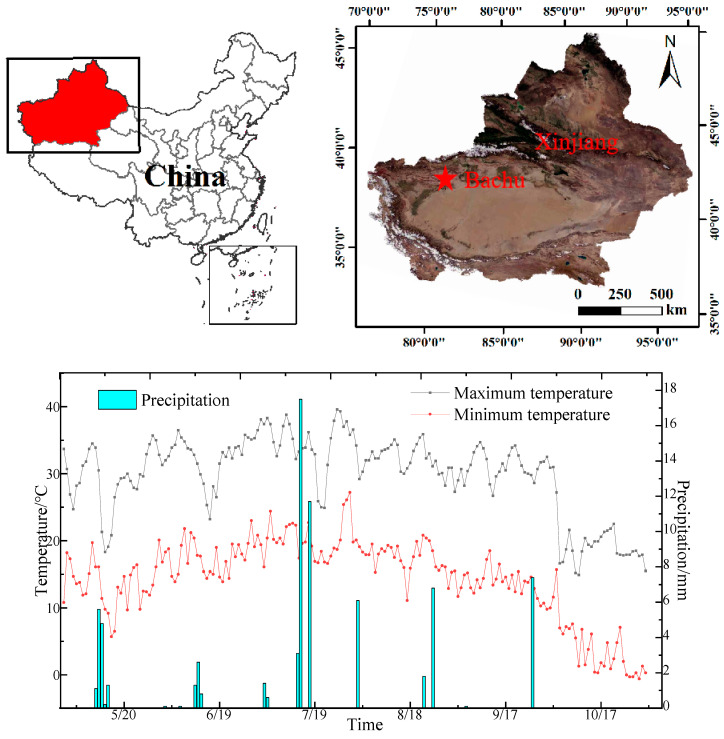
Meteorological data and geographic location of the study area.

**Figure 5 plants-12-01383-f005:**
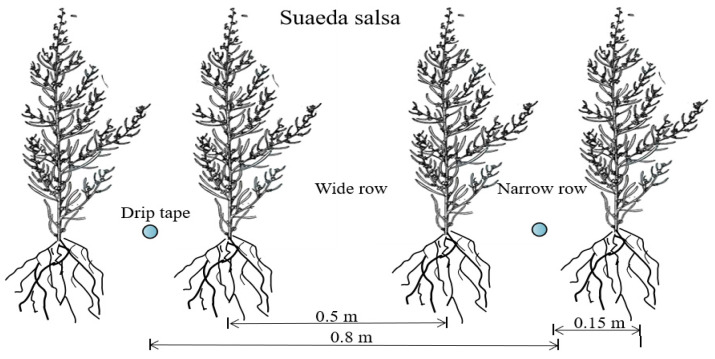
Planting schematics of drip-irrigated *Suaeda salsa*.

**Table 1 plants-12-01383-t001:** Effects of different planting densities and irrigation volumes on the plant height of *Suaeda salsa* under drip irrigation.

Treatment	Seedling Stage	Mature Stage	Blooming Stage	Fruiting Stage
W1D1	23.52 ± 2.11 d	59.26 ± 2.81 d	80.26 ± 1.05 e	85.20 ± 2.36 f
W1D2	27.49 ± 1.02 c	68.26 ± 2.13 c	95.23 ± 0.92 c	108.63 ± 3.06 d
W1D3	25.69 ± 1.21 c	56.63 ± 0.95 e	76.25 ± 1.99 e	99.36 ± 5.63 e
W1D4	20.35 ± 1.63 d	50.36 ± 3.69 e	71.20 ± 2.63 e	82.36 ± 4.65 f
W2D1	28.33 ± 1.23 c	70.32 ± 3.63 c	99.02 ± 3.26 c	102.52 ± 7.96 d
W2D2	34.78 ± 1.63 b	85.99 ± 2.62 b	120.63 ± 3.32 b	135.09 ± 4.75 b
W2D3	36.22 ± 1.02 b	83.16 ± 2.33 b	119.52 ± 3.15 b	142.52 ± 6.56 b
W2D4	25.13 ± 0.69 c	61.52 ± 1.26 d	87.82 ± 2.97 d	100.25 ± 3.85 d
W3D1	25.22 ± 1.03 c	62.35 ± 1.97 d	88.36 ± 3.52 d	90.69 ± 3.63 e
W3D2	31.74 ± 0.98 c	79.52 ± 2.63 b	111.06 ± 3.66 b	126.52 ± 4.21 c
W3D3	45.98 ± 2.03 a	96.52 ± 2.99 a	136.63 ± 3.16 a	164.52 ± 8.69 a
W3D4	26.99 ± 1.22 c	67.13 ± 3.22 c	95.03 ± 5.63 c	108.52 ± 4.52 d
F value				
W	184.66 **	376.31 **	50.51 **	916.71 **
D	166.31 **	229.40 **	50.02 **	1944.04 **
W × D	25.38 **	36.43 **	13.31 **	122.91 **

Note: Values in the same column followed by different lowercase letters are significantly different at the *p* ≤ 0.05 probability level. W represents irrigation water volume; D represents density; W × D represents water and density interaction; ** Indicates that in there was significant difference in *p* ≤ 0.01 level.

**Table 2 plants-12-01383-t002:** Effect of different planting densities and irrigation volumes on the stem diameter of *Suaeda salsa* under drip irrigation.

Treatment	Seedling Stage	Mature Stage	Blooming Stage	Fruiting Stage
W1D1	2.59 ± 0.05 b	9.15 ± 0.12 a	16.26 ± 0.63 a	25.63 ± 1.96 a
W1D2	2.28 ± 0.11 b	6.23 ± 0.52 b	10.36 ± 0.99 b	12.36 ± 1.52 c
W1D3	2.01 ± 0.22 b	4.52 ± 0.63 c	5.02 ± 0.92 d	8.00 ± 1.20 d
W1D4	1.42 ± 0.24 c	2.49 ± 0.56 d	2.69 ± 0.35 e	2.99 ± 0.37 e
W2D1	3.26 ± 0.11 a	9.26 ± 1.25 a	17.20 ± 0.65 a	30.29 ± 2.62 a
W2D2	3.11 ± 0.15 a	6.69 ± 0.63 b	16.32 ± 0.26 a	20.00 ± 0.55 b
W2D3	2.11 ± 0.02 b	6.09 ± 0.62 b	9.12 ± 0.88 c	13.63 ± 0.23 c
W2D4	1.65 ± 0.14 c	3.02 ± 0.26 d	4.16 ± 0.33 e	5.63 ± 0.55 e
W3D1	3.01 ± 0.23 a	9.24 ± 0.52 a	16.69 ± 2.63 a	27.63 ± 4.63 a
W3D2	2.51 ± 0.08 b	6.52 ± 0.52 b	12.36 ± 1.26 b	17.36 ± 1.11 b
W3D3	2.22 ± 0.05 b	5.36 ± 0.36 c	7.26 ± 0.85 c	11.63 ± 1.06 c
W3D4	1.52 ± 0.04 c	2.78 ± 0.15 d	3.03 ± 0.63 e	4.55 ± 0.25 e
F value				
W	25.61 **	11.43 **	223.26 **	184.23 **
D	142.75 **	546.81 **	3023.95 **	2368.29 **
W × D	4.68 **	2.51 *	48.84 **	13.42 **

Note: Values in the same column followed by different lowercase letters are significantly different at the *p* ≤ 0.05 probability level. W represents irrigation water volume; D represents density; W × D represents water and density interaction; * indicates a significant difference at the level of *p* ≤ 0.05; ** Indicates that in there was significant difference in *p* ≤ 0.01 level.

**Table 3 plants-12-01383-t003:** Effect of different planting densities and irrigation volumes on the canopy width of *Suaeda salsa* under drip irrigation.

Treatment	Seedling Stage	Mature Stage	Blooming Stage	Fruiting Stage
W1D1	15.39 ± 1.63 a	33.95 ± 2.36 a	42.63 ± 2.69 a	44.62 ± 1.65 a
W1D2	12.36 ± 0.26 b	20.26 ± 0.56 b	25.63 ± 0.63 b	31.42 ± 1.36 b
W1D3	4.68 ± 0.12 c	6.96 ± 0.26 c	9.69 ± 0.33 c	13.69 ± 1.52 c
W1D4	3.26 ± 0.33 d	4.63 ± 0.52 d	5.06 ± 0.15 d	6.52 ± 0.59 d
W2D1	20.23 ± 2.58 a	38.63 ± 3.68 a	46.36 ± 4.56 a	46.63 ± 5.26 a
W2D2	17.52 ± 1.06 b	22.52 ± 1.63 b	26.52 ± 2.63 b	33.06 ± 3.26 b
W2D3	7.59 ± 0.58 c	10.96 ± 0.55 c	19.63 ± 1.66 c	22.63 ± 2.21 c
W2D4	4.63 ± 0.17 d	7.26 ± 0.36 d	8.76 ± 0.52 d	10.96 ± 0.14 d
W3D1	17.63 ± 1.58 a	39.52 ± 3.23 a	45.52 ± 3.55 a	46.23 ± 3.69 a
W3D2	16.53 ± 0.89 b	22.62 ± 1.63 b	26.16 ± 2.58 b	31.23 ± 2.62 b
W3D3	5.96 ± 0.11 c	13.56 ± 0.52 c	16.23 ± 1.23 c	20.23 ± 1.36 c
W3D4	4.23 ± 0.05 d	6.96 ± 0.22 d	7.36 ± 0.23 d	8.72 ± 0.15 d
F value				
W	133.15 **	34.38 **	95.20 **	29.91 **
D	1422.48 **	998.29 **	3521.10 **	4980.69 **
W × D	10.42 **	2.24 Ns	16.69 **	37.92 **

Note: Values in the same column followed by different lowercase letters are significantly different at the *p* ≤ 0.05 probability level. W represents irrigation water volume; D represents density; W × D represents water and density interaction; ** Indicates that in there was significant difference in *p* ≤ 0.01 level; Ns indicates no significant difference (*p* > 0.05).

**Table 4 plants-12-01383-t004:** Effect of different planting densities and irrigation volumes on the biomass of *Suaeda salsa* under drip irrigation.

Treatment	Seedling Stage	Mature Stage	Blooming Stage	Fruiting Stage
W1D1	5.22 ± 0.42 b	14.04 ± 0.88 c	15.84 ± 0.69 d	17.92 ± 0.85 c
W1D2	4.98 ± 0.36 c	14.23 ± 0.52 c	15.91 ± 0.41 d	16.84 ± 0.15 d
W1D3	4.82 ± 0.23 c	13.24 ± 0.52 d	14.46 ± 0.85 e	15.67 ± 0.66 e
W1D4	4.38 ± 0.42 c	12.43 ± 0.27 e	13.15 ± 0.25 f	13.82 ± 0.52 f
W2D1	4.97 ± 0.33 c	14.63 ± 0.36 c	15.78 ± 0.16 d	17.11 ± 0.33 c
W2D2	6.19 ± 0.46 a	17.01 ± 0.88 a	19.79 ± 0.99 a	21.94 ± 1.06 a
W2D3	5.94 ± 0.15 a	16.86 ± 0.79 a	18.81 ± 0.88 b	20.21 ± 0.69 b
W2D4	4.92 ± 0.16 c	14.24 ± 0.23 c	14.91 ± 0.52 d	15.66 ± 0.25 e
W3D1	5.02 ± 0.42 c	14.73 ± 0.16 c	16.79 ± 0.16 c	17.99 ± 0.12 c
W3D2	5.68 ± 0.43 b	16.04 ± 0.52 b	18.09 ± 0.26 b	19.82 ± 0.99 b
W3D3	5.42 ± 0.22 b	16.57 ± 0.64 b	18.73 ± 0.99 b	21.17 ± 0.46 b
W3D4	4.82 ± 0.36 c	14.02 ± 0.43 c	15.62 ± 0.42 d	16.74 ± 0.25 d
F value				
W	30.55 **	16.08 **	39.31 **	17.21 **
D	58.88 **	73.34 **	95.26 **	97.20 **
W × D	5.82 **	1.84 Ns	7.00 **	5.16 **

Note: Values in the same column followed by different lowercase letters are significantly different at the *p* ≤ 0.05 probability level. W represents irrigation water volume; D represents density; W × D represents water and density interaction; ** Indicates that in there was significant difference in *p* ≤ 0.01 level; Ns indicates no significant difference (*p* > 0.05).

**Table 5 plants-12-01383-t005:** Effect of different planting densities and irrigation volumes on the ash content and salt uptake of drip-irrigated *Suaeda salsa*.

Treatment	Seedling Stage		Mature Stage		Blooming Stage		Fruiting Stage	
	Ash Content (g·kg^−1^)	Salt Uptake (kg·hm^−2^)	Ash Content (g·kg^−1^)	Salt Uptake (kg·hm^−2^)	Ash Content (g·kg^−1^)	Salt Uptake (kg·hm^−2^)	Ash Content (g·kg^−1^)	Salt Uptake (kg·hm^−2^)
W1D1	354.26 ± 12.00 a	1749.22 ± 67.97 a	382.33 ± 2.94 a	5369.43 ± 632.32 a	323.51 ± 7.01 a	5123.05 ± 364.39 a	308.67 ± 2.06 a	5540.65 ± 105.13 a
W1D2	327.02 ± 2.98 b	1628.57 ± 50.95 b	343.29 ± 7.05 a	4884.29 ± 271.69 b	290.48 ± 12.30 a	4622.16 ± 486.76 ab	277.28 ± 12.85 b	4669.46 ± 484.64 bc
W1D3	308.07 ± 10.91 c	1484.90 ± 67.88 c	331.76 ± 14.95 a	4392.44 ± 34.96 c	280.68 ± 10.94 a	4058.68 ± 136.96 b	267.95 ± 5.09 b	4198.84 ± 179.94 c
W1D4	296.92 ± 9.07 c	1300.51 ± 15.78 d	319.14 ± 12.06 b	3967.16 ± 213.68 d	270.05 ± 20.99 b	3551.70 ± 235.45 c	257.80 ± 18.97 b	3562.80 ± 210.61 d
W2D1	332.66 ± 9.45 b	1652.66 ± 15.73 b	351.64 ± 11.78 a	5145.13 ± 36.97 a	299.39 ± 2.42 a	4725.59 ± 88.73 a	290.50 ± 9.93 a	4970.48 ± 89.71 b
W2D2	300.06 ± 11.81 c	1857.96 ± 91.89 a	321.91 ± 3.91 b	5476.32 ± 114.41 a	270.55 ± 20.95 b	5354.79 ± 344.57 a	268.64 ± 18.15 b	5894.02 ± 294.56 a
W2D3	292.32 ± 18.02 c	1736.39 ± 39.23 a	316.39 ± 9.50 b	5333.13 ± 149.50 a	257.67 ± 3.73 b	4846.82 ± 51.54 a	264.07 ± 21.98 b	5336.81 ± 462.35 ab
W2D4	283.51 ± 18.90 d	1394.87 ± 83.40 d	307.58 ± 1.71 b	4378.71 ± 54.86 c	253.41 ± 8.02 b	3778.85 ± 50.11 c	248.37 ± 21.00 b	3889.47 ± 88.95 d
W3D1	303.93 ± 11.90 c	1525.75 ± 68.88 c	314.44 ± 17.78 b	4631.14 ± 160.81 b	266.34 ± 11.03 b	4471.85 ± 66.00 b	254.22 ± 9.00 b	4573.40 ± 57.78 c
W3D2	272.37 ± 6.04 d	1547.06 ± 58.05 c	285.74 ± 13.48 c	4584.36 ± 303.37 b	241.93 ± 4.83 c	4376.52 ± 60.75 b	230.91 ± 7.82 c	4573.55 ± 67.66 c
W3D3	258.66 ± 2.97 e	1402.98 ± 91.01 cd	273.76 ± 13.13 c	4535.62 ± 219.38 b	231.92 ± 17.95 c	4344.28 ± 251.02 b	221.34 ± 3.89 c	4685.77 ± 73.82 bc
W3D4	254.59 ± 18.90 e	1228.14 ± 144.42 e	269.66 ± 13.24 c	3780.63 ± 163.08 d	228.52 ± 3.79 c	3569.48 ± 31.20 c	217.92 ± 6.53 c	3647.98 ± 299.87 d
*p*-Value								
W	44.58 **	15.35 **	13.60 **	18.26 **	25.10 **	10.15 **	33.78 **	17.20 **
D	8.03 *	19.15 **	9.05 **	27.96 **	10.37 *	16.46 **	13.35 *	35.02 **
W × D	3.21 **	4.65 **	2.17 *	2.33 Ns	4.26 **	4.27 **	6.16 **	3.66 *

Note: Values in the same column followed by different lowercase letters are significantly different at the *p* ≤ 0.05 probability level. W represents irrigation water volume; D represents density; W × D represents water and density interaction; * indicates a significant difference at the level of *p* ≤ 0.05; ** Indicates that in there was significant difference in *p* ≤ 0.01 level; Ns indicates no significant difference (*p* > 0.05).

**Table 6 plants-12-01383-t006:** Effect of different planting densities and irrigation volumes on the WUE and WSE of *Suaeda salsa* under drip irrigation.

Treatment	WUE (kg·m^−3^)	WSE (kg·m^−3^)
W1D1	5.98 ± 0.10 a	1.85 ± 0.09 a
W1D2	5.61 ± 0.34 a	1.56 ± 0.09 b
W1D3	5.22 ± 0.13 b	1.40 ± 0.04 c
W1D4	4.61 ± 0.07 c	1.19 ± 0.08 d
W2D1	4.56 ± 0.09 c	1.33 ± 0.08 d
W2D2	5.85 ± 0.04 a	1.57 ± 0.10 b
W2D3	5.39 ± 0.05 b	1.42 ± 0.02 c
W2D4	4.18 ± 0.28 e	1.04 ± 0.12 e
W3D1	4.00 ± 0.09 d	1.02 ± 0.06 e
W3D2	4.40 ± 0.22 c	1.02 ± 0.05 e
W3D3	4.70 ± 0.01 c	1.04 ± 0.01 e
W3D4	3.72 ± 0.22 d	0.81 ± 0.05 f
F value		
W	274.32 **	261.75 **
D	102.01 **	45.50 **
W × D	10.45 **	8.78 **

Note: Values in the same column followed by different lowercase letters are significantly different at the *p* ≤ 0.05 probability level. W represents irrigation water volume; D represents density; W × D represents water and density interaction; ** Indicates that in there was significant difference in *p* ≤ 0.01 level.

**Table 7 plants-12-01383-t007:** Multiple regression relationships between irrigation, density, and each evaluation indicator.

Dependent Variable Y	Regression Equation	R^2^	*p*≤
Plant height/Y1	Y1 = −4.13 × 10^−5^W^2^ + 0.30W + 4.16 × 10^−4^WD−0.17D^2^ + 14.22D − 774.42	0.79	0.01
Stem diameter/Y2	Y2 = −6.43 × 10^−6^W^2^ + 0.051W − 1.79 × 10^−5^WD + 0.011D^2^ − 1.72D − 27.83	0.97	0.01
Canopy width/Y3	Y3 = −5.31 × 10^−6^W^2^ + 0.039W + 5.67 × 10^−5^WD + 0.0095D^2^ − 2.31D + 30.88	0.99	0.01
Ash content/Y4	Y4 = −2.38 × 10^−5^W^2^ + 0.13W + 2.90 × 10^−4^WD − 0.039D^2^ − 6.00D + 235.86	0.98	0.01
Biomass/Y5	Y5 = −2.41 × 10^−6^W^2^ + 0.016W + 8.18 × 10^−5^WD − 0.015D^2^ + 0.96D − 32.17	0.77	0.01
Salt uptake/Y6	Y6 = −0.0011W^2^ + 6.67W + 0.025WD − 2.65D^2^ + 101.92D − 8336.11	0.80	0.01
WUE/Y7	Y7 = −3.82 × 10^−7^W^2^ + 9.12 × 10^−4^W + 2.64 × 10^−5^WD − 0.0034D^2^ + 0.19D + 1.42	0.86	0.01
WSE/Y8	Y8 = −1.84 × 10^−7^W^2^ + 5.73 × 10^−4^W + 1.02 × 10^−5^WD − 6.42 × 10^−4^D^2^ + 0.0069D + 1.13	0.97	0.01

The letters W and D denote the amount of irrigation water and density used, respectively. Each binary quadratic regression equation was established based on the least squares method.

**Table 8 plants-12-01383-t008:** Maximum theoretical values for each evaluation indicator and the corresponding irrigation and density values.

Y	Ymax	W	D
Biomass/Y5	20.88	4163.27	43.47
Salt uptake/Y6	5547.60	3612.24	36.12
WUE/Y7	5.91	3000.00	39.18
WSE/Y8	1.74	3000.00	30.00

The letters W and D denote the amount of irrigation water and density used, respectively.

**Table 9 plants-12-01383-t009:** Physical and chemical properties of the soil.

Soil Texture	Soil Bulk Density (g·cm^−3^)	Soil Salt Content (g·kg^−1^)	Field Capacity (%)	Organic Matter (g·kg^−1^)	Available Phosphorus (mg·kg^−1^)	Available Potassium (mg·kg^−1^)	Available Nitrogen (mg·kg^−1^)
Sandy loam	1.46	25.63	23.75	10.32	8.63	133.66	54.68

**Table 10 plants-12-01383-t010:** Experimental design.

Irrigation Volume (m^3^·hm^−2^)	Density (Plants·m^−2^)	Treatment
3000	30	W1D1
3000	40	W1D2
3000	50	W1D3
3000	60	W1D4
3750	30	W2D1
3750	40	W2D2
3750	50	W2D3
3750	60	W2D4
4500	30	W3D1
4500	40	W3D2
4500	50	W3D3
4500	60	W3D4

**Table 11 plants-12-01383-t011:** Suaeda salsa growth period and irrigation plan.

Growth Period	Seedling Stage	Mature Stage	Blooming Stage	Fruiting Stage	Full Growth Period
Data	5.10~6.19	6.20~8.10	8.11~9.10	9.11~10.10	5.10~10.10
Days/day	40	51	30	30	151
Irrigation times	4	5	3	2	14
W1/(m^3^·hm^−2^)	200	240	200	200	3000
W2/(m^3^·hm^−2^)	225	300	250	300	3750
W3/(m^3^·hm^−2^)	250	360	300	400	4500

## Data Availability

Not applicable.

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
