# Peer review of "Interactive Effect of Irrigation Volume and Planting Density on Growth and Salt Uptake in Field-Grown Drip-Irrigated *Suaeda salsa* (L.) Pall."

_plants, 2023, doi:10.3390/plants12061383_

Round 1

Reviewer 1 Report

The Manuscript focuses on the effects of irrigation and plant density on the Suaeda salsa, growing on saline soil and salt uptake. Generally, the manuscript is well-organized and clear. Below I report some comments:

Please report the authorship for the Latin name in the title and in the abstract as well as in the main text when it is cited for the first time

Please specify in the introduction the choice of the Suaeda salsa

In the material and methods please report the references for the methods used for soil properties, especially for salt content and for plants. Please also specify the range values of salt content for saline soil with a reference in table 1.

In table 3, please specify the meaning of “data”.

Line 139 Do you mean water use efficiency or water productivity?

Please report the p-values in each tables

Please improve figure 3, it is difficult to read letters

Reviewer 2 Report

This is very practically-oriented study with low scientific novelty and relevance. Among study limitations, no actual measurement of salinity-associated ions in soil was performed before and after the experiment, and no chemical components of salts accumulated in plants were measured. The main benefit could be the creation of an optimization model for watering and planting density, but this does not take into account the effects of climate and soil properties, so it is not generalizable.

Title

Not comprehensive. Absolutely not clear what is "irrigation quota". 

Something like this: "Interactive effect of irrigation volume and planting density on growth and salt uptake in field-grown drip-irrigated Suaeda salsa".

Abstract

Replace "irrigation quota" with "irrigation intensity" here and throughout the manuscript.

Replace "growth statistics" by "growth characteristics" (line 17).

Very much doubt if herbs has something to be called "crown" (line 20), is it "shoot width"?

Line 21–22, "appropriate planting density" is a suggestion made by authors, not measured parameter.

It needs to be explained what is meant by "salt absorption" (line 24 and further). If it is accumulated content, it cannot much decrease with time. 

Introduction

It is necessary to introduce the model species and to characterize what is known about its salinity tolerance and salt accumulation ability.

Literature survey on previous studies on effects of irrigation and planting density performed with Suaeda salsa does not give the impression that another study is needed. It is only logical that actual amount of irrigated water needs to be balanced with plant needs, depending also on natural precipitation and soil properties, therefore, differences in results. What other conclusion could theoretically be made here?

Materials and methods

Use past tense when describing performed actions.

It seems to be problematic to use parameter "water salt transfer efficiency" because salts were taken by plants from soil not from water" (line 141). What is meant by this?

Include full names of stages in column headings for Table 3. Indicate dates in a comprehensive manner (May 10, June 19 etc.). It is indicated that growth period stopped on October 10, but sampling at fruiting stage was performed on October 15, what happened during these 5 days?

Replace "baked" with "dried" (line 132).

The standard parameter WUE means "water use efficiency", it has nothing to do with "water production".

Results

Do not include sentences from discussion with references (lines 155–157) and from discussion (lines 314–317).

Include full names of stages in column headings for Tables 4, 5, 6, 7 and legend to Figure 3.

In all notes to tables and figures, replace "alphabets" with "letters".

Color scheme used in Figure 3 do not allow for easy perception and comparison. No explanation is included what is meant by "W:30.55**" etc. on the graph.

Discussion is very limited in respect to analysis of the real reasons for the differences in results from different studies, assuming that background climate and soil conditions are irrelevant. Otherwise, the obtained results is said to be "similar to the results" from other studies, making one wonder once more why this study was necessary.

List of references is extremely flawed and disordered.

Round 2

Reviewer 2 Report

Thank you, all comments have been properly addressed